# Genetics of Generalized Pustular Psoriasis: Current Understanding and Implications for Future Therapeutics

**DOI:** 10.3390/genes14061297

**Published:** 2023-06-20

**Authors:** Syuan-Fei Yang, Min-Huei Lin, Pei-Chen Chou, Sheng-Kai Hu, Sin-Yi Shih, Hsin-Su Yu, Sebastian Yu

**Affiliations:** 1Department of Dermatology, Kaohsiung Medical University Hospital, Kaohsiung Medical University, Kaohsiung 807, Taiwan; ggemstone02@gmail.com (S.-F.Y.); yup.kmu@gmail.com (H.-S.Y.); 2School of Medicine, College of Medicine, Kaohsiung Medical University, Kaohsiung 807, Taiwan; u110001147@gap.kme.edu.tw (M.-H.L.); u110001005@gap.kmu.edu.tw (P.-C.C.); thekingdoghu@gmail.com (S.-K.H.); isabel890711@gmail.com (S.-Y.S.); 3Graduate Institute of Clinical Medicine, College of Medicine, Kaohsiung Medical University, Kaohsiung 807, Taiwan; 4Department of Dermatology, College of Medicine, Kaohsiung Medical University, Kaohsiung 807, Taiwan; 5Neuroscience Research Center, Kaohsiung Medical University, Kaohsiung 807, Taiwan

**Keywords:** psoriasis and genetics, pustular psoriasis, inflammatory skin disease, autoimmunity, IL36RN, CARD14, AP1S3, MPO, SERPINA1, SERPINA3

## Abstract

Psoriasis is a chronic inflammatory skin disease characterized by the appearance of clearly demarcated erythematous and scaly plaques. It can be divided into various types, including plaque, nail, guttate, inverse, and pustular psoriasis. Plaque psoriasis is the most commonly occurring type, though there is another rare but severe pustular autoinflammatory skin disease called generalized pustular psoriasis (GPP), which manifests with acute episodes of pustulation and systemic symptoms. Though the etiopathogenesis of psoriasis is not yet fully understood, a growing body of literature has demonstrated that both genetic and environmental factors play a role. The discovery of genetic mutations associated with GPP has shed light on our comprehension of the mechanisms of the disease, promoting the development of targeted therapies. This review will summarize genetic determinants as known and provide an update on the current and potential treatments for GPP. The pathogenesis and clinical presentation of the disease are also included for a comprehensive discussion.

## 1. Introduction

Psoriasis is a common chronic inflammatory skin disease with a variety of clinical manifestations [1]. Psoriasis may be classified into non-pustular and pustular forms. Pustular psoriasis may be further stratified into localized and generalized forms [2]. It is believed that both environmental and genetic factors participate in the immune mechanisms of psoriasis [3]. Current studies have demonstrated genetic susceptibility to psoriasis involving components of both innate and adaptive immune systems [1]. Prolonged inflammation results in dysregulated keratinocyte proliferation and differentiation, and the keratinocytes participate in both the initiation and maintenance phases of psoriasis [4].

Psoriasis vulgaris (PV) is known to be the most common subtype of psoriasis. Both immune and genetic studies have identified interleukin (IL)-23 and IL-17 as the main drivers of psoriasis vulgaris [5,6]. It is characterized by relatively stable and localized erythematous scaly plaques. On the other hand, pustular psoriasis (PP) is rarer but potentially life-threatening and is associated with innate immune system overactivation. It may present with erythematous, scaly skin, including pustules and systemic neutrophilia. Pustular psoriasis can present in various forms, including localized pustules, as in acrodermatitis continua of Hallopeau (ACH) or palmoplantar pustulosis (PPP), or diffuse, non-acral pustules with systemic inflammation, as in generalized pustular psoriasis (GPP) [2].

GPP is a severe type of psoriatic disease. It is characterized by the onset of widespread, macroscopically visible pustules on non-acral skin with or without systemic symptoms such as fever, neutrophilia, and elevated serum levels of C-reactive protein [7]. The extent of systemic symptoms varies among patients as well as between flares within the same patient.

Clarifying the immune mechanisms behind GPP helps to develop potential therapeutic targets for this disease. Meanwhile, we should also keep in mind that the age of onset and the frequency of genetic mutations vary significantly among different subtypes [8].

In 2017, Akiyama et al. first proposed the term “autoinflammatory keratinization diseases” (AiKDs) to describe the inflammatory keratinization of the skin due to genetic autoinflammatory pathomechanisms [9]. As the pathogenic mechanism of AiKD becomes elucidated, there will be more appropriate treatment methods and precision medicines available [10]. This novel concept also sheds light on the development of therapeutic agents for pustular psoriasis.

Recent studies of the molecular pathomechanisms of pustular psoriasis suggest that the inhibition of specific cytokines, including the IL-36 axis, is a potential therapeutic strategy to control the disease activity of pustular psoriasis [11].

Autoimmunity is characterized by the activation of the adaptive immune system, including T and B cells, while autoinflammatory responses are driven by endogenous danger signals as well as inflammatory mediators and cytokines. In complex inflammatory conditions such as psoriasis, these two processes frequently coexist and can influence and trigger each other. This review will discuss the mechanism of psoriasis based on the autoimmune and autoinflammatory processes that are activated. We also aim to provide an up-to-date elucidation of the genetic mutations associated with different subtypes of pustular psoriasis and, ultimately, focus on biological treatments available for GPP.

## 2. Genetics of Pustular Psoriasis

Although the first GPP case was reported a century ago, its etiology and detailed pathogenesis have only been discussed within the last ten years (Table 1). It was not until 2011 that IL36RN was initially discovered as a gene responsible for causing GPP [12,13]. Since then, a growing number of genetic mutations such as CARD14, AP1S3, MPO, and the SERPIN family have been identified as associated with GPP. However, not all GPP patients carry mutations of these genes, suggesting that there are still other genetic factors to be discovered. These disease-causing genes may participate in common or similar pathogenic molecular pathways [14].

Ethnic differences in GPP should also take into consideration. For example, pathogenic mutations of AP1S3 have been reported in individuals of European origin but not in Malaysian populations [15,16], while MPO and SERPINA3 variants were identified in patients of European descent [17,18]. Associations with other ethnic groups remain to be elucidated.

The cases of pustular psoriasis are classified into GPP, PPP, and ACH according to the ERASPEN criteria [2]. Assan et al. suggested that PPP and ACH might be separate diseases while still maintaining some overlap [19]. Accordingly, there are prospective phenotype–genotype and multi-omics studies to better recognize the mechanisms of each subgroup. Another study conducted in Italy in a real-life setting revealed the concomitant rate of plaque psoriasis, which was the greatest in GPP and the least in ACH [20]. To distinguish GPP alone from those with PV is quite important since the selection of treatment is based on the disease mechanism and the clinical phenotype, which can include GPP alone, ACH alone, predominate ACH, ACH evolving into GPP, and ACH with GPP.

Adult-onset immunodeficiency syndrome (AOID) is known as an AIDS-like illness with abnormal interferon-γ (IFN-γ)/IL12 signaling. It is associated with high-titer neutralizing antibodies to IFN-γ, the controller of numerous pathogens [21]. The majority of cases exhibit skin-related symptoms, such as reactive skin conditions (82%) and infectious skin diseases (45%), with neutrophilic dermatoses being the most common among them [21,22]. A recent study conducted by Piranit et al. supports that both GPP and AOID involving pustular reactions are diseases caused by dysregulated proteolytic and apoptotic processes [23]. Clinically and genetically, GPP and AOID are likely to share some common pathogenetic mechanisms. To date, there have been no reports of AOID and GPP occurring in the same individuals or within the same families. However, genetic research has found heterozygous variants in the SERPINA3 and SERPINA1 genes in patients with AOID and GPP, respectively [24,25].

### 2.1. IL36RN

IL-36 cytokines are relatively novel and belong to **the** IL-1 family, which has members that are produced by many sources, such as epithelial cells, myeloid dendritic cells, and monocytes. IL36RN encodes for IL-36Ra, which inhibits the pro-inflammatory effects of IL-36 cytokines by binding their receptors, then preventing the release of mediators that stimulate the pustule formation seen in GPP [26].

Onoufriadis et al. reported that IL-36RN mutations can cause sporadic GPP, and according to their study, IL-36 mutations underline sporadic European GPP, as well as Tunisian autosomal recessive GPP [12]. Additionally, the first Asian case of GPP associated with IL36RN mutations was reported in 2012, therefore indicating that IL36RN mutations are common in some GPP cases worldwide [27]. The prevalence of IL36RN mutations among pustular psoriasis subtypes is different; patients with GPP have the highest prevalence of these mutations (23.7%). This is followed by ACH, which has the second-highest prevalence (17.4%), and lastly, PPP demonstrates the lowest prevalence of these mutations (5.1%) [8].

Hence, in order to ascertain if IL36RN alleles are the crucial determinants of pustular psoriasis across various disease subtypes, a regression analysis was carried out, incorporating clinical diagnosis as a covariate [28]. Individuals with homozygous mutations of IL36RN tend to experience more severe disease manifestations compared to those with heterozygous mutations, and these mutations are inherited through an autosomal recessive pattern [29]. Another study indicated that IL36RN mutations are almost not seen in individuals with both PPP and GPP [30]. Accordingly, this finding suggests that a large proportion of cases of GPP alone are caused by homozygous or compound heterozygous mutations of IL36RN.

On the other hand, the presence of IL36RN disease alleles demonstrated a dose-dependent influence on the age of onset across all types of pustular psoriasis [28]. According to genetic analyses, the frequency of IL36 mutations plays a role in differentiating pustular psoriasis subtypes [8]. Sophie et al. found that the percentage of individuals carrying IL36RN disease-associated alleles was higher in those with GPP and ACH. Individuals with GPP and ACH were more likely to have biallelic mutations compared to those affected by PPP.

### 2.2. CARD14

Caspase recruitment domain family member 14 (CARD14) is a gene located in the psoriasis susceptibility locus 2 (PSORS2). CARD is a protein-binding molecule that facilitates the formation of complexes containing CARD proteins, which are involved in apoptosis and NF-κB signaling pathways. Among them, CARD14 is found to be specifically expressed in diseases of the skin and is primarily localized in the basal and suprabasal epidermal layers [31]. Some CARD proteins are related to chronic inflammatory skin diseases, such as early-onset sarcoidosis or amyopathic dermatomyositis [32]. The role of CARD14 mutations as either causal factors or disease susceptibility factors for PV, GPP, or pityriasis rubra pilaris may depend on the specific mutation or variant position within the CARD14 gene. [28].

Differences in ethnical groups and geographic areas affect the outcome to some extent. A study revealed that the carrier rate of the CARD14 variant in Japanese individuals is higher than in Europeans. Therefore, we can consider CARD14 an important predisposing factor for GPP with PV in the Japanese population [33,34].

### 2.3. AP1S3

The AP1S3 gene, which encodes adaptor protein complex 1 (AP-1), plays a crucial role in stabilizing AP-1 heterotetramers that participate in vesicular trafficking between the trans-Golgi network and endosomes [35]. Cells with mutations in AP1S3 have decreased autophagosome formation in keratinocytes, leading to p62 build-up and resulting in enhanced NF-κB signaling [16]. Loss-of-function mutations of the AP1S3 gene were found relevant in GPP, which implies pustular psoriasis as an autoinflammatory manifestation resulting from impaired vesicular trafficking [15].

The pathogenic variants are distributed mainly in Europeans and rarely in East Asians and Africans. The variant frequency of AP1S3 in GPP patients of European ancestry is about 10.8% [15]. Suppressing AP1S3 expression in human keratinocytes and HEK293 cells eliminates endosomal activation by polyinosinic-polycytidylic acid, a TLR3 agonist involved in responding to viral infections. Researchers suggested that abnormalities in vesicular trafficking could be a significant pathological basis for the autoinflammatory process in pustular psoriasis [15].

Another study investigating genetic variations in patients with pustular psoriasis found that AP1S3 mutations were in fewer GPP cases than IL36RN, and patients with AP1S3 disease alleles were mainly female [8].

### 2.4. MPO

Deficiencies in MPO, a heme-containing peroxidase secreted by neutrophil granulocytes that catalyzes the formation of reactive oxygen species (ROS), have just been identified in association with GPP [14]. The association between MPO deficiency and pustular skin disease was first recognized by Vergnano et al. with phenome-wide association studies [36], and in vitro functional studies showed that mutations in the MPO gene lead to elevated neutrophil accumulation and activity, suggesting a role of MPO mutations in the pathogenesis of GPP [37].

The quantity of mutant MPO alleles was positively correlated with a younger age of onset, which is similar to the genotype-phenotype correlation of the IL36RN gene and further validates the genetic correlation of GPP [17]. The discovery that the MPO gene plays a pathogenic role in GPP provides perspectives on understanding GPP pathogenesis.

### 2.5. SERPINA1, SERPINA3

SERPINA1 and SERPINA3 are **i**nhibitors of cathepsin G, the primary serine protease involved in cleaving and activating IL-36 precursors. The loss of function of these protease inhibitors may induce severe inflammatory effects [25]. Additionally, heterozygous loss-of-function mutations in both SERPINA1 and SERPINA3 were identified in individuals with GPP, and decreased protease inhibitor activity may result in enhanced IL-36 activation [18].

A study conducted by Piranit et al. reinforced the concept that the biological functions of SERPINB3 involve inhibiting cysteine proteases when mutated, and the subsequent overactivation of proteases leads to an intensified inflammatory reaction accompanied by heightened neutrophil recruitment [23]. Patients carrying SERPINB3 mutations exhibit aberrant SERPINB3 expression. The accumulation of misfolded SERPINB3 proteins causes the overactivation of cathepsin L, followed by the inactivation of SERPINA1, finally evolves into AOID with pustular reactions [38,39].

### 2.6. BTN3A3

BTN3A3 belongs to the human butyrophilin (BTN) 3 family, which has the ability to activate the NF-κB pathway, resulting in an excessive inflammatory response by suppressing the expression of IL-36Ra. To investigate the molecular pathogenesis of GPP, Q. Zhang et al. conducted a whole-exome sequencing study in the Chinese Han population [40]. However, the result found only two loci identified with exome-wide significance: the strongest one was in the IL36RN gene, and the other was located within the MHC region. A subsequent gene burden test demonstrated a correlation between BTN3A3 and GPP. Subtype analysis revealed that both IL36RN and BTN3A3 were markedly linked to GPP alone and GPP with PV. The BTN3A3 gene carried two LOF mutations with the most significant association. As a previously unreported determinant of GPP, BTN3A3 acted as a key regulator of cell proliferation, and its expression was associated with inflammatory imbalance.

### 2.7. TGFBR2

TGF-β signaling is recognized for its inhibitory effects on cell proliferation and immune system suppression [41]. Thus, the hyperproliferation of keratinocytes in the psoriatic epidermis is consistent with disrupted TGF-β signaling because of heterozygous loss-of-function TGFBR2 mutations. Concomitant with the overexpression of KRT17, there is an increase in keratinocyte proliferation and subsequent recruitment of neutrophils [42]. The overexpression of KRT17 is thus in line with a potential role for diminished TGFBR2 function in both GPP and AOID. Whole-exome sequencing (WES) was carried out on a total of 53 patients, comprising 32 individuals exhibiting pustular psoriasis phenotypes and 21 individuals with AOID presenting with pustular skin reactions [43]. The result showed that 4 Thai patients displaying similar pustular phenotypes, including two diagnosed with GPP and two with AOID, were found to carry the same rare TGFBR2 frameshift mutation. It is concluded that AOID might share pathogenic mechanisms with GPP.

Mechanistically, TGFBR1 and TGFBR2 are transmembrane serine/threonine kinases [44]. TGFBR2 expression is remarkably reduced or absent in psoriatic skin. As a result, it has been suggested that genetic variations in TGFBR2 could enhance susceptibility to GPP and AOID in some patients.

## 3. Current and Potential Therapeutic Agents Targeting Immune Mediators in Generalized Pustular Psoriasis

The phenotype and pathogenesis of different psoriasis subtypes are on a spectrum. On the one hand, plaque psoriasis is associated with the overactivation of the adaptive immune system, including T and B cells, and is thought to involve self-perpetuating inflammatory mechanisms through the IL-23/Th17 axis [45]. On the other end, pustular psoriasis has been associated with the stimulation of innate immune responses and the activation of IL-36 cytokine pathways [46]. Based on the pathomechanism, therapeutic agents for patients who have plaque psoriasis and GPP at the same time need to target not only the adaptive immune pathways but also the innate immune axis [47].

IL-36 cytokines are members of the IL-1 superfamily, and the IL-1/IL-36–chemokine–neutrophil axis plays a significant role in driving disease pathology in GPP. The first pathogenic variant found to be linked with GPP was a homozygous mutation of the IL36RN gene [48], and further studies have looked into the distribution over different populations [48,49].

Progress in understanding the relationship between autoinflammation and clinical phenotypes has contributed to the development of highly efficacious targeted treatments such as TNF-α, IL-17, IL-23, IL-1α/β, or IL-36 inhibitors or receptor blockers, as well as small molecule drugs such as PDE4 inhibitors, JAK inhibitors, and ROR-γt inhibitors.

Well-established treatment guidelines for GPP are currently lacking, and multiple biologic and non-biologic treatments exist. Considering the variety of comorbidities and severity associated with GPP, personalized treatments should be tailored. Figure 1 shows a graphical abstract of current and emerging biologic agents for GPP.

### 3.1. IL-36 Pathway Inhibitors

Anti-IL-36 receptor antibodies can be employed to block the signaling pathway responsible for GPP flares and can be effective for patients with mutant IL36RN [48].

At present, only a single GPP-specific treatment, spesolimab, an interleukin-36 receptor antagonist, has received approval for use in the United States. With the experience of GPP complete remission after two doses of spesolimab [50], spesolimab was then approved by European Commission in adult GPP flares [51]. Spesolimab has been demonstrated to reduce the levels of relevant serum biomarkers and cellular populations in the skin lesions of patients with GPP, such as CD3+ T, CD11c+, and IL-36γ+ cells and lipocalin-2-expressing cells.

In patients with GPP, spesolimab has been observed to induce rapid changes in commonly disrupted molecular pathways in both GPP and PPP, suggesting that it may have the potential to improve clinical outcomes [52]. The results of a randomized controlled trial indicated that a 900 mg intravenous infusion of spesolimab led to greater lesion resolution in a patient group experiencing active GPP flare-ups after one week [53]. The improvement of the condition was evaluated using the GPPGA, which is a standardized assessment of a subject’s skin status based on three factors: erythema, pustules, and scaling/crusting [54]. After one week, there were almost four times the number patients who received spesolimab and achieved a GPPGA total score of 0 or 1 compared to control patients. Furthermore, it was found that spesolimab may relate to a higher incidence of infection, though neither opportunistic nor severe [55]. Long-term management options also were assessed; patient-reported outcomes were improved, and markers of systemic inflammation were normalized [56]. Recent research also indicates that spesolimab is effective for patients with GPP without IL36RN mutations [57].

Additional potential therapies targeting the IL-36 pathway for GPP are currently under development. Imsidolimab, an IL-36 inhibitor, recently passed through a phase 3 clinical trial to evaluate its efficacy and safety [58]. Patients received 750 mg of IV imsidolimab on day 1 and added 100 mg of subcutaneous imsidolimab every 4 weeks until day 85. Imsidolimab exhibited a rapid and sustained alleviation of symptoms and pustular eruptions in patients with GPP.

There are currently efforts underway to develop small molecule inhibitors of IL-36γ, which could have the potential to treat GPP. A-552 was identified as a potent inhibitor of IL-36γ in humans [59]. Phenotypic analysis of individuals without the IL-36R-encoding gene disclosed that they do not exhibit severe immunodeficiency, further supporting that the IL-36 pathway is a promising therapeutic target with minimal side effects [60].

### 3.2. IL-1RAcP

Interleukin-1 receptor accessory protein (IL-1RAcP) antibodies represent another feasible treatment alternative for GPP patients. IL-1RAcP, a member of the immunoglobulin superfamily proteins, has a crucial function in the signaling of the IL-1 family cytokines, such as IL-1, IL-33, and IL-36. Blocking IL-1RAcP’s ability to form a dimer with IL-36R could prevent the overactivation of the IL-36 pathway and subsequent inflammation [61]. Zarezadeh et al. indicated IL-1RAcP as a potential therapeutic target for inflammatory and autoimmune diseases [62]. However, the long-term safety and effectiveness of IL-1RAcP antibodies need to be determined since IL-1RAcP is expressed in a wide range of cell types, and excessive suppression may result in multiple toxic effects [63].

### 3.3. TNF-**α** Inhibitors

TNF-***α***, produced by activated plasmacytoid dendritic cells (DCs) and damaged keratinocytes, can stimulate the IL-36 pathway. TNF-***α*** inhibitors indirectly suppress the expression of IL-36γ, resulting in reduced activation of the pro-inflammatory IL-36 pathway [64]. Adalimumab, infliximab, and certolizumab pegol are TNF-***α*** inhibitors that have been approved for GPP treatment in Japan [65]. Cases with rapid and sustained resolution of skin lesions after infliximab used were reported in Poland [66]. A retrospective study showed the treatment efficacy rate of pustule clearance, which was 100% in the adalimumab + acitretin group [67].

However, paradoxical GPP is a potential adverse effect of TNF-***α*** inhibitors. A study conducted in Turkey involving 156 GPP patients revealed that TNF-***α*** inhibitors were the only biologic that triggered paradoxical GPP [68]. It is estimated that 0.6%-5.3% of patients receiving TNF-***α*** inhibitors developed paradoxical GPP, with infliximab being the most frequently associated biologic with this condition [69].

### 3.4. IL-17 Inhibitors

Secukinumab, ixekizumab, and brodalumab are biologics that have been proven to manage GPP patients in Japan [70,71,72]. A retrospective study in Germany compared the rate of excellent response to GPP patients, with 60.0% in the secukinumab group and 50.0% in the ixekizumab group [73]. A phase IV, multicenter, open-label randomized control trial in Japan demonstrated that skin lesions mostly resolved in GPP patients under ixekizumab treatment, and there were no side effects reported [74].

However, there was a case of a Japanese individual that developed increased serum levels of liver enzymes during treatment with brodalumab for generalized pustular psoriasis [75]. The relationship between brodalumab and autoimmune hepatitis (AIH)/primary biliary cholangitis (PBC) overlap syndrome should be noted.

These IL-17 inhibitors mentioned above have been shown to be effective in controlling flares in the acute phase or as maintenance therapy in adult patients with GPP.

### 3.5. IL-23 Inhibitors

IL-23 regulates the production of IL-17, which subsequently stimulates the synthesis of pro-inflammatory IL-36R agonists, leading to the overactivation of the IL-36 pathway. The IL-23 inhibitors risankizumab and guselkumab are indicated for the treatment of GPP in Japan [76,77]. Ustekinumab, as an IL-12/23 antagonist, has been introduced to GPP patients, who achieved complete remission after its dose was titrated [78]. Additionally, both newly diagnosed ACH cases and already known therapy-refractory ACH cases had satisfactory and sustained therapy responses to guselkumab and risankizumab [79].

This suggests that IL-23 inhibitors may control flares in the acute phase or as maintenance therapy in adult patients with GPP.

### 3.6. Additional Biological Therapy and Non-Biologic Options

While the TNF-***α***/IL-17/IL-23 axis is predominantly targeted in plaque psoriasis, the IL-1/IL-36–chemokine–neutrophil axis shows greater potential as a therapeutic target in GPP. Previous studies have explored the use of IL-1 targeting biologics, such as the IL-1***α*** receptor antagonist anakinra, as well as the IL-1***β***βmonoclonal antibodies gevokizumab and canakinumab, in GPP patients [80]. Anakinra is a successful treatment in patients with GPP carrying mutant IL36RN genes, while gevokizumab and canakinumab are effective in blocking the pro-inflammatory cytokine IL-1β [81].

As for non-biologic immunomodulatory management, methotrexate, cyclosporine, apremilast, and retinoids have been used for the treatment of GPP, but the efficacy is only based on case reports and non-randomized studies. Japanese guidelines have suggested the use of topical treatments as maintenance therapy following flares or as a supplementary therapy to address psoriasis-like symptoms [65].

## 4. Conclusions

GPP is a severe inflammatory disease distinct from PV [82]. Recent genetic observations and investigations provided us with insight into the disease. We found specific genes that are associated with pustular skin disease, including IL36RN, CARD14, AP1S3, MPO, SERPINA1, SERPINA3, BTN3A3, and TGFBR2. The immunologic pathway implicates IL-36 as a central node cytokine. That is, GPP constitutes a large IL-36-dominated keratinocyte cytokine storm and epidermal neutrophil aggregation.

The advances in our comprehension of GPP and its treatment options have the potential to improve patient care. It is known that the IL-36 pathway is the main inflammatory pathway implicated in GPP, but it is neither necessary nor sufficient to cause the disease. Aside from genes that play a role in the regulation of IL-36 signaling, there are IL36RN-negative GPP cases that have been noted. Due to the rarity of GPP, it has been challenging to identify additional disease-causing genes in the past. However, by combining whole-exome sequence data from various centers and targeting cases that are more prone to be monogenic in origin, progress could be achieved.

Progressive biologic therapies that target different chemokine receptors show efficacy, but there are also safety considerations. As more relevant and efficacious treatment options become available, patient outcomes and quality of life will improve. We should also keep in mind that immediate treatment goals during GPP flares are to alleviate skin inflammation and minimize the impact of systemic symptoms to avoid complications such as cardiovascular aseptic shock, heart failure, acute respiratory distress syndrome, prerenal kidney failure, neutrophilic cholangitis, uveitis, and severe infections [80,83]. Prevention of flare-ups of GPP is another treatment goal, and further clinical studies are indicated to evaluate the efficacy of the prevention of GPP flares. Additionally, the prevalence of genetic mutations of GPP varies in different countries and ethnic groups. It is important to investigate if patients with different genetic mutations of GPP have different short-term and long-term treatment responses.

## Figures and Tables

**Figure 1 genes-14-01297-f001:**
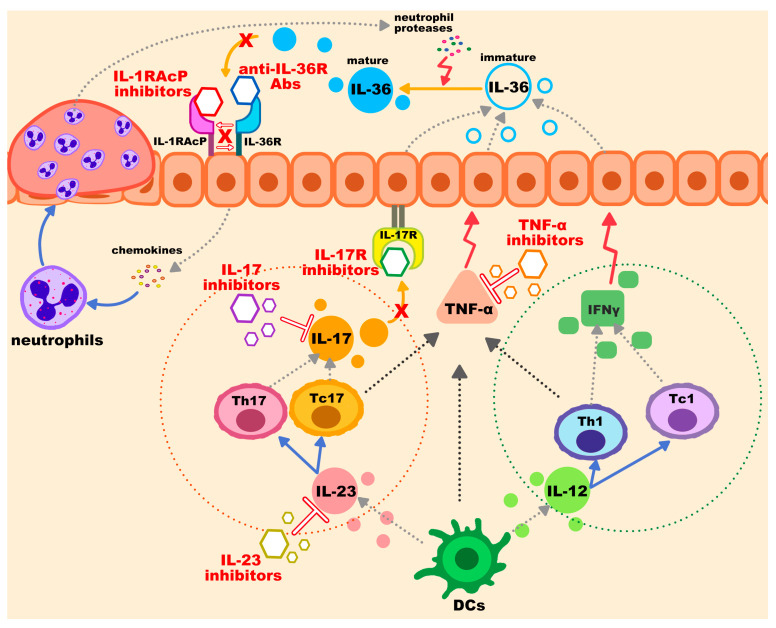
Graphical abstract of mechanisms of current and potential biologic agents for generalized pustular psoriasis.

**Table 1 genes-14-01297-t001:** Genetic mutations associated with generalized pustular psoriasis and their proposed effects.

Gene	Protein	Proposed Effect	Proposed GPP Mechanism
*IL36RN*	Interleukin-36 receptor antagonist	LOF	Loss of IL-36 antagonism
*CARD14*	Caspase recruitment domain family member 14	GOF	Enhanced NF-*κ*B signaling
*AP1S3*	Adaptor-related protein complex 1 subunit sigma 3	LOF	Enhanced NF-*κ*B signaling
*MPO*	Myeloperoxidase	LOF	Enhanced neutrophil protease activity; decreased neutrophil turnover
*SERPINA1*	Serpin family A member 1	LOF	Loss of protease inhibitor activity; enhanced IL-36 activation
*SERPINA3*	Serpin family A member 3	LOF	Loss of protease inhibitor activity; enhanced IL-36 activation
*BTN3A3*	Butyrophilin subfamily 3 member A3	LOF	Enhanced NF-*κ*B signaling; disturbed IL-36 inflammatory axis
*TGFBR2*	Transforming growth factor β receptor 2	LOF	Stimulated keratinocytes to produce cytokines

LOF: loss-of-function. GOF: gain-of-function.

## Data Availability

No new data were created or analyzed in this study. Data sharing is not applicable to this article.

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
