# Peer review of "Genetics of Generalized Pustular Psoriasis: Current Understanding and Implications for Future Therapeutics"

_genes, 2023, doi:10.3390/genes14061297_

Round 1

Reviewer 1 Report

Dear authors,

Is an excellent paper that present a comprehensive review about genetic mutations associated with GPP and the current and potential biologic therapies for GPP. I appreciate the detailed analysis of each gene mutations and the therapeutic options for this particular form of psoriasis. 

Author Response

Response to Reviewer 1

Dear authors,

Is an excellent paper that present a comprehensive review about genetic mutations associated with GPP and the current and potential biologic therapies for GPP. I appreciate the detailed analysis of each gene mutations and the therapeutic options for this particular form of psoriasis. 

Response: Thank you for your comments.

Reviewer 2 Report

The authors have presented a review article about the genetics of GPP. The authors collected information on the genetics of GPP and provide an update on the current and potential treatments for GPP. Major problems: 

  1. As is well known GPP, despite is the type of psoriasis, it a is a distinct entity from plaque psoriasis (different cytokines, different genetics, different pathogenesis). In my opinion if the authors dedicate the article GPP it should be focused only on GPP (besides basic information in introduction). In this review tere is a mix information of genetics / pathogenetic aspects of both - plaque psoriasis and GPP, which is really confusing. 
  2. Some of cited articles (even in genetic part of article) refer to plaque psoriasis not GPP and taking into consideration the fact that genetics of psoriasis is entity different from GPP, the information about genetics of psoriasis vulgarism shouldn’t be placed in this part. 
  3. In general article for me is quite chaotic. In introduction is the information about pathogenesis of disease are limited and should be more detailed as is important as a matter of topic. Also lines 69-90 and part 2.1 are a little bit confusing. In my opinion article should be re-organized. 
  4. In my opinion the title of article doesn’t suit to the main part - genetics aspect of GPP is 3 pages and the rest article is about treatment etc. 
  5. The confusion should be improved and summarize a little bit the main genetics aspects. 
  6. The review lacks of a methodology section: methods for collection and selection (included criteria for inclusion and exclusion of articles used for the review) of data are not described (also lack of prisma)
  7. To improve article authors should consider add table / chart etc. 

Author Response

Response to Reviewer 2

The authors have presented a review article about the genetics of GPP. The authors collected information on the genetics of GPP and provide an update on the current and potential treatments for GPP. Major problems: 

  1. As is well known GPP, despite is the type of psoriasis, it a is a distinct entity from plaque psoriasis (different cytokines, different genetics, different pathogenesis). In my opinion if the authors dedicate the article GPP it should be focused only on GPP (besides basic information in introduction). In this review tere is a mix information of genetics / pathogenetic aspects of both - plaque psoriasis and GPP, which is really confusing. 

Response: Thank you for your comments. We have revised the manuscript to focus on GPP. Also, we have deleted HLA-C*06:02 paragraph as this allele is related to psoriasis vulgaris but not GPP.

  1. Some of cited articles (even in genetic part of article) refer to plaque psoriasis not GPP and taking into consideration the fact that genetics of psoriasis is entity different from GPP, the information about genetics of psoriasis vulgarism shouldn’t be placed in this part. 

Response: Thank you for your comments. We have revised the manuscript. In this version, we give general information about psoriasis in the introduction but emphasize GPP is distinct from psoriasis vulgaris. In the genetics and treatments sections, we focus on GPP and cited references which are relevant to GPP.

  1. In general article for me is quite chaotic. In introduction is the information about pathogenesis of disease are limited and should be more detailed as is important as a matter of topic. Also lines 69-90 and part 2.1 are a little bit confusing. In my opinion article should be re-organized. 
    Response: Thank you for your comments. We have re-organized the introduction. We have deleted 2.1 AOID paragraph. Part of the content of AOID that is relevant to GPP is rephrased and integrated into the genetics section.
  2. In my opinion the title of article doesn’t suit to the main part - genetics aspect of GPP is 3 pages and the rest article is about treatment etc. 

Response: Thank you for your comments. We have changed the title of the article to “Genetics of generalized pustular psoriasis: current understanding and implications for future therapeutics” as this review article encompasses not only genetics but also current and emerging therapeutics derived from current understanding of genetic mutations.

  1. The confusion should be improved and summarize a little bit the main genetics aspects. 
    Response: Thank you for your suggestions. We have added a Table in this revised version to summarize genetic mutations of GPP and proposed biologic

effects of these mutations

  1. The review lacks of a methodology section: methods for collection and selection (included criteria for inclusion and exclusion of articles used for the review) of data are not described (also lack of prisma)
    Response: Thank you for your comments. Because this article is a narrative review but not systematic review, we didn’t adopt PRISMA, which is primarily for systematic review and meta-analysis. All literature relevant to GPP was searched in PubMed, Embase, and Google Scholar. The literature retrieved on these search engines was reviewed by the authors and relevant content was integrated by the authors. See Author Contributions section.
  2. To improve article authors should consider add table / chart etc. 

Response: Thank you for your suggestions. In this revised version, we have added a table to summarized genetic mutations of GPP and added a figure to summarize treatments for GPP.

Reviewer 3 Report

 The review “Genetics of generalized pustular psoriasis” by Yang et al., summarizes the genetic determinants of generalized pustular psoriasis (GPP) and provides an update on the current and potential treatments for GPP. The authors also included pathogenesis and clinical presentation of the disease.  The review is interesting and very well written with lot of information.

The authors described at length the disease-causing gene of GPP and the association of the disease with different races and age groups. The request the authors to include mechanism of actions of the genes/mutations and how they are linked with the disease.

The authors also narrated the current and potential therapeutic agents targeting immune mediators in generalized pustular psoriasis. Potential of few blockers and inhibitors like IL-36 pathway inhibitors and IL-17 inhibitors has been discussed. I request the authors to include the mechanism of action of the inhibitors in the form of diagram/charts which will be very interesting and easy to understand for the readers.

Overall, the review provides useful information on genetics of generalized pustular psoriasis and potential treatment options. The manuscript narrates relevant advancement on GPP and may be very useful for researchers working on the topic.

English is great except few typos.

Author Response

Response to Reviewer 3

The review “Genetics of generalized pustular psoriasis” by Yang et al., summarizes the genetic determinants of generalized pustular psoriasis (GPP) and provides an update on the current and potential treatments for GPP. The authors also included pathogenesis and clinical presentation of the disease.  The review is interesting and very well written with lot of information.
Response: Thank you for your comments.

The authors described at length the disease-causing gene of GPP and the association of the disease with different races and age groups. The request the authors to include mechanism of actions of the genes/mutations and how they are linked with the disease.
Response: Thank you for your suggestions. We have added a table to summarized the genetic mutations of GPP and the proposed biologic effects of these mutations.

The authors also narrated the current and potential therapeutic agents targeting immune mediators in generalized pustular psoriasis. Potential of few blockers and inhibitors like IL-36 pathway inhibitors and IL-17 inhibitors has been discussed. I request the authors to include the mechanism of action of the inhibitors in the form of diagram/charts which will be very interesting and easy to understand for the readers.
Response: Thank you for your suggestions. We have added a figure to illustrate mechanisms of current and emerging biologic agents for GPP. 

Overall, the review provides useful information on genetics of generalized pustular psoriasis and potential treatment options. The manuscript narrates relevant advancement on GPP and may be very useful for researchers working on the topic.
Response: Thank you for your comments.